# Front-end Weber-Fechner gain control enhances the fidelity of combinatorial odor coding

Nirag Kadakia[1], Thierry Emonet[1,2]*

[1]Department of Molecular, Cellular and Developmental Biology, Yale University, New Haven, United States; [2]Department of Physics, Yale University, New Haven, United States

**Abstract** We showed previously (Gorur-Shandilya et al., 2017) that *Drosophila* olfactory receptor neurons (ORNs) expressing the co-receptor Orco scale their gain inversely with mean odor intensity according to Weber-Fechner's law. Here, we show that this front-end adaptation promotes the reconstruction of odor identity from dynamic odor signals, even in the presence of confounding background odors and rapid intensity fluctuations. These enhancements are further aided by known downstream transformations in the antennal lobe and mushroom body. Our results, which are applicable to various odor classification and reconstruction schemes, stem from the fact that this adaptation mechanism is not intrinsic to the identity of the receptor involved. Instead, a feedback mechanism adjusts receptor sensitivity based on the activity of the receptor-Orco complex, according to Weber-Fechner's law. Thus, a common scaling of the gain across Orco-expressing ORNs may be a key feature of ORN adaptation that helps preserve combinatorial odor codes in naturalistic landscapes.

DOI: https://doi.org/10.7554/eLife.45293.001

*For correspondence:
thierry.emonet@yale.edu

## Introduction

Animals identify and discriminate odors using olfactory receptors (Ors) expressed in olfactory receptor neurons (ORNs) (*Joseph and Carlson, 2015*; *Buck and Axel, 1991*; *Clyne et al., 1999*; *Vosshall et al., 1999*). Individual ORNs, which typically express a single Or, respond to many odorants, while individual odorants activate many distinct ORNs (*Friedrich and Korsching, 1997*; *Hallem and Carlson, 2006*; *Wang et al., 2010*; *Nara et al., 2011*). Odors are thus encoded by the combinatorial patterns of activity they elicit in the sensing periphery (*Malnic et al., 1999*; *Wang et al., 2010*; *Hildebrand and Shepherd, 1997*; *Hallem and Carlson, 2006*; *de Bruyne et al., 2001*; *Friedrich and Korsching, 1997*), and these patterns are decoded downstream into behavioral response (*Wilson, 2013*; *Davies et al., 2015*). Still, ethologically relevant odors are often mixed with background ones (*Saha et al., 2013*; *Renou et al., 2015*) and intensity can vary widely and rapidly as odors are carried by the wind (*Murlis et al., 1992*; *Weissburg, 2000*; *Celani et al., 2014*; *Cardé and Willis, 2008*). How are odors recognized reliably despite these confounds? In *Drosophila melanogaster*, ORN dose response curves exhibit similar Hill coefficients but distinct power-law distributed activation thresholds (*Hallem and Carlson, 2006*; *Si et al., 2019*), which together with inhibitory odorants enhance coding capacity (*Si et al., 2019*; *Cao et al., 2017*; *Hallem and Carlson, 2006*; *Stevens, 2016*). In antennal lobe (AL) glomeruli, mutual lateral inhibition normalizes population response, reducing the dependency of activity patterns on odor concentration (*Asahina et al., 2009*; *Olsen et al., 2010*). Further downstream, sparse connectivity to the mushroom body (MB) helps maintain neural representations of odors, and facilitates compressed sensing and associative learning schemes (*Caron et al., 2013*; *Litwin-Kumar et al., 2017*; *Krishnamurthy et al., 2017*;

*Dasgupta et al., 2017*). Finally, temporal features of neural responses contribute to concentration-invariant representations of odor identity (*Brown et al., 2005*; *Raman et al., 2010*; *Gupta and Stopfer, 2014*; *Wilson et al., 2017*).

Here, we examine how short-time ORN adaptation at the very front-end of the insect olfactory circuit contributes to the fidelity of odor encoding. Our theoretical study is motivated by the recent discovery of invariances in the signal transduction and adaptation dynamics of ORNs expressing the co-receptor Orco. ORN response is initiated upon binding of odorant molecules to olfactory receptors (ORs), opening the ion channels they form with the co-receptor Orco (*Larsson et al., 2004*; *Butterwick et al., 2018*). Because of differences in odor-receptor affinities, the responses of ORNs to diverse odorants of the same concentration differ widely (*Hallem and Carlson, 2006*; *Montague et al., 2011*; *Stensmyr et al., 2012*). In contrast, downstream from this input nonlinearity, signal transduction and adaptation dynamics exhibit a surprising degree of invariance with respect to odor-receptor identity: reverse-correlation analysis of ORN response to fluctuating stimuli produces highly stereotyped, concentration-invariant response filters (*Martelli et al., 2013*; *Si et al., 2019*; *Gorur-Shandilya et al., 2017*).

These properties stem in part from an apparently invariant adaptive scaling law in ORNs: gain varies inversely with mean odor concentration according to the Weber-Fechner Law of psychophysics (*Weber, 1996*; *Fechner, 1860*), irrespective of the odor-receptor combination (*Gorur-Shandilya et al., 2017*; *Cafaro, 2016*; *Cao et al., 2016*). This invariance can be traced back to adaptative feedback mechanisms in odor transduction, upstream of ORN firing (*Nagel and Wilson, 2011*; *Cao et al., 2016*; *Cafaro, 2016*; *Gorur-Shandilya et al., 2017*), which depend on the activity of the signaling pathway rather than on the identity of its receptor (*Nagel and Wilson, 2011*). The generality of the adaptive scaling suggests it could be mediated by the highly conserved Orco co-receptor (*Butterwick et al., 2018*; *Getahun et al., 2013*; *Getahun et al., 2016*; *Guo et al., 2017*), which has been already been implicated in other types of odor adaptation, taking place over longer timescales (*Guo and Smith, 2017*; *Guo et al., 2017*).

While in a simpler system such as *E. coli* chemotaxis (*Waite et al., 2018*), adaptive feedback via the Weber-Fechner Law robustly maintains sensitivity over concentration changes, the implication for a multiple-channel system – which combines information from hundreds of cells with overlapping receptive fields – is less clear. Here, we combine a biophysical model of ORN adaptive response and neural firing with various sparse signal decoding frameworks to explore how ORN adaptation with Weber-Fechner scaling affects combinatorial coding and decoding of odor signals spanning varying degrees of intensity, molecular complexity, and temporal structure. We find that this front-end adaptive mechanism promotes the accurate discrimination of odor signals from backgrounds of varying molecular complexity, and aids other known mechanisms of neural processing in the olfactory circuit to maintain representations of odor identity across environmental changes.

## Results

### Model of ORN sensing repertoire

To model ORN firing rates in response to time-dependent odor signals, we extended a minimal model (*Gorur-Shandilya et al., 2017*) that reproduces the Weber-Fechner gain adaptation and firing rate dynamics measured in individual *Drosophila* ORNs in response to Gaussian and naturalistic signals (code available on GitHub, *Kadakia, 2019*; copy archived at https://github.com/elifesciences-publications/ORN-WL-gain-control).

We consider a repertoire of $M = 50$ ORN types that each express one type of Or together with the co-receptor Orco (*Larsson et al., 2004*). Within ORNs of type $a = 1, ..., M$, Or-Orco complexes form non-selective cation channels (*Butterwick et al., 2018*) (*Figure 1A*) that switch between active and inactive conformations, while simultaneously binding to odorants $i$ with affinity constants, $K_{ai}^*$ and $K_{ai}$, respectively (*Nagel and Wilson, 2011*; *Gorur-Shandilya et al., 2017*). For simplicity, we only consider agonists, that is $K_{ai}^* > K_{ai}$, and assume receptors can only bind one odorant at a time. The analysis can easily be extended to include inhibitory odorants, which increases coding capacity (*Cao et al., 2017*). Dissociation (inverse affinity) constants are chosen from a power law distribution

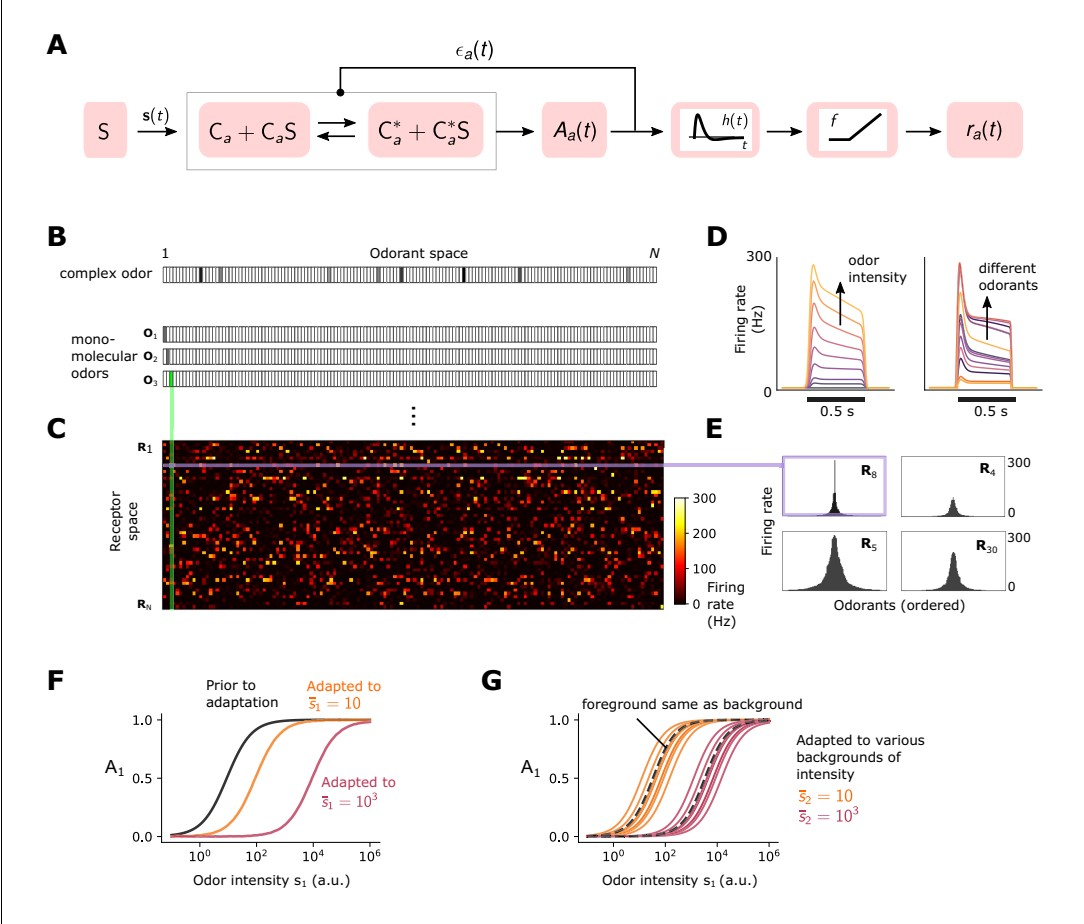

**Figure 1.** Simple ORN model (*Gorur-Shandilya et al., 2017*). (A) Or/Orco complexes of type $a$ switch between active $C_a^*$ and inactive conformations $C_a$. Binding an exitatory odorant (S in the diagram) favors the active state. The active fraction is determined by the free energy difference between inactive and active conformations of the Or/Orco complex in its unbound state, $\epsilon_a(t)$ (in units of $k_B T$), and by odorant binding with affinity constants $\mathbf{K}_a^* = (K_{a1}^*, ..., K_{ai}^*, ..., K_{aN}^*)$ and $\mathbf{K}_a$ for the active and inactive conformations, respectively (*Equations 1-2*). Adaptation is mediated by a negative feedback (*Nagel and Wilson, 2011*) from the activity of the channel onto the free energy difference $\epsilon_a(t)$ with timescale $\tau$. ORN firing rates $r_a(t)$ are generated by passing $A_a(t)$ through a linear temporal filter $h(t)$ and a nonlinear thresholding function $f$. (B) Odors are represented by $N$-dimensional vectors $\mathbf{s} = [s_1, ..., s_i, ..., s_N]$, whose components $s_i$ are the concentrations of the individual molecular constituents of $\mathbf{s}$. (C) Step-stimulus firing rate of 50 ORNs to the $N$=150 possible monomolecular odorants $\mathbf{s} = s_i$, given power-law distributed afffinity constants (*Si et al., 2019*). (D) Temporal responses of a representative ORNs to a pulse stimulus, for a single odorant at several intensities (left), or to many odorants of the same intensity (right). (E) Representative ORN tuning curves (a single row of the response matrix in C, ordered by magnitude). Tuning curves are diverse, mimicking measured responses (*Hallem and Carlson, 2006*). (F) Dose-response of an ORN before (black) and after adaptation to either a low (yellow) or high (magenta) odor concentration. (G) Same, but the ORN was allowed to first adapt to one of various backgrounds of differing identities, before the foreground (same as in F) was presented. Also shown is the specific case when the foreground and background have the same identity (dashed lines).

DOI: https://doi.org/10.7554/eLife.45293.002

($\alpha = 0.35$) recently found across ORN-odor pairs in *Drosophila* larvae (*Si et al., 2019*). For a handful of ORNs, we choose a very large value for one of the $K_{ai}^*$ to mimic high responders to private odorants relevant to innate responses (*Stensmyr et al., 2012*). These private odors do not affect the general findings.

Assuming that odorant binding and conformation changes are faster than other reactions in the signaling pathway, the fraction of channels of type $a$ that are active at steady state is:

$$A_a(t) = \frac{C_a^* + C_a^* \mathbf{K}_a^* \cdot \mathbf{s}(t)}{C_a^* + C_a^* \mathbf{K}_a^* \cdot \mathbf{s}(t) + C_a + C_a \mathbf{K}_a \cdot \mathbf{s}(t)}. \tag{1}$$

$C_a$ and $C_a^*$ represent unbound channels in the inactive and active conformation. Here,

$\mathbf{K}_a \cdot \mathbf{s}(t) = \sum_i^N K_{ai} s_i(t)$, where $s_i(t)$ is the time-dependent concentration of the $i$-th monomolecular component of the odor signal $\mathbf{s}(t)$ at time $t$ (**Figure 1B**). $N = 150$ is the size of the molecular odorant space (**Figure 1B**). **Equation 1** can be rearranged as (derivation in Materials and methods):

$$A_a(t) = \left[ 1 + \exp\left( \epsilon_a(t) + \ln\left( \frac{1 + \mathbf{K}_a \cdot \mathbf{s}(t)}{1 + \mathbf{K}_a^* \cdot \mathbf{s}(t)} \right) \right) \right]^{-1}. \tag{2}$$

The two terms in the exponential represent the change in the channel's free energy due to the binding of odorant $i$, and the free energy difference $\epsilon_a$ between the unbound states $C_a$ and $C_a^*$, in units of $k_B T$. Because $K_{ai}^* > K_{ai}$, a sudden increase in the concentration of excitatory odor results in an increase in activity $A_a$.

Upon prolonged stimulation, ORNs adapt. At least one form of adaptation, which takes place over short time scale, $\tau \simeq 250~\mathrm{ms}$ (**Gorur-Shandilya et al., 2017**), involves a negative feedback of the Or-Orco channel activity onto the channel sensitivity (**Nagel and Wilson, 2011**; **Gorur-Shandilya et al., 2017**). To model this adaptation process, we assume that inward currents elicited by activating Or-Orco channels eventually result in an increase of the free energy difference $\epsilon_a(t)$, possibly via a feedback onto Orco (**Butterwick et al., 2018**):

$$\tau \frac{d\epsilon_a(t)}{dt} = A_a(t) - A_{0a}, \tag{3}$$

where $\epsilon_{L,a} < \epsilon_a(t) < \epsilon_{H,a}$. The lower bound $\epsilon_{L,a}$ determines the spontaneous activity of the channel. The higher bound $\epsilon_{H,a}$ determines the concentrations of odors at which adaptation is unable to keep up and saturation occurs (**Gorur-Shandilya et al., 2017**). Through these dynamics, $\epsilon_a(t)$ can compensate for changes in free energy due to ligand binding (see **Equation 2**), returning the activity $A_a$ towards an adapted level $A_{0a}$ above the spontaneous activity. Since $\epsilon_a$ is bounded below, a minimum amount of signal intensity is needed for adaptation to kick in. Finally, the firing rate is modeled by passing the activity $A_a(t)$ through the derivative-taking bi-lobed filter $h(t)$ and a rectifying nonlinearity $f$ (**Gorur-Shandilya et al., 2017**):

$$r_a(t) = f(h(t) \otimes A_a(t)), \tag{4}$$

where $\otimes$ is convolution. When deconvolved from stimulus dynamics, the shapes of the temporal kernels of *Drosophila* ORNs that express Orco tend to be stereotyped for many odor-receptor combination (**Martelli et al., 2013**; **Gorur-Shandilya et al., 2017**; **Si et al., 2019**) (although there are known exceptions such as super-sustained responses **Montague et al., 2011**). Moreover, adaptation is not intrinsic to the receptor (**Nagel and Wilson, 2011**). Accordingly, for simplicity $\tau$, $h(t)$, and $f$ are assumed independent of receptor and odorant identities.

This minimal model reproduces the essential features of ORN response to odorant pulses (**Nagel and Wilson, 2011**; **Martelli et al., 2013**; **Cao et al., 2016**). In the absence of stimulus, ORNs fire spontaneously at rates (1–10 Hz) (**Hallem and Carlson, 2006**) set by the lower free energy bound $\epsilon_{L,a}$, which we choose from a normal distribution (**Figure 1D**). For sufficiently strong stimuli, adaptation causes $\epsilon_a$ to increase, compensating for the drop in free energy difference due to ligand binding. This gradually reduces the firing rate to a steady state level $r(A_{0a}) \simeq$ 30–40 Hz (**Gorur-Shandilya et al., 2017**) (**Figure 1D**). The diversity of temporal firing responses and tuning curves measured experimentally (**Hallem and Carlson, 2006**; **Montague et al., 2011**; **Brown et al., 2005**; **Gupta and Stopfer, 2014**; **Raman et al., 2010**) arise naturally in the model due to the distribution of chemical affinity constants and the nonlinearity of **Equation 2** (**Figure 1B-Figure 1E**).

The model also reproduces Weber-Fechner scaling of the gain with the inverse of the mean odorant intensity $\bar{s}_i$ (**Gorur-Shandilya et al., 2017**; **Cao et al., 2016**). For small fluctuations $\Delta s_i$ around $\bar{s}_i$, we have from **Equation 2** that $\Delta A_a / \Delta s_i \simeq A_a(\bar{s}_i)(1 - A_a(\bar{s}_i))/\bar{s}_i$, whereby Weber's Law is satisfied provided $A_a(\bar{s}_i)$ is approximately constant (derivation in Materials and methods). In our model, since the rate of adaptation depends only on the activity of the ion channel (right hand-side of **Equation 3**), then in the adapted state we have $A_a(\bar{s}_i) \simeq A_{0a}$, ensuring that the gain scales like $1/\bar{s}_i$. This process adjusts the sensitivity of the ORN by matching the dose responses to the mean signal concentration, while maintaining their log-slopes (**Figure 1F**). However, for foreground odors mixed with

background odors to which the system has adapted, the dose response curves now exhibit background-dependent shifts (*Figure 1G*).

While this phenomenological model could be extended to include further details – for example, we could relax the quasi-steady-state assumption in *Equation 2*, use a more complex model for channel adaptation and neural firing (*Gorur-Shandilya et al., 2017*), or consider feedforward mechanisms in addition to negative integral feedback (*Schulze et al., 2015*) – this minimally parameterized form captures the key dynamical properties of Orco-expressing ORNs relevant to our study: receptor-independent adaptation (*Nagel and Wilson, 2011*) with Weber-Fechner scaling (*Gorur-Shandilya et al., 2017*; *Cafaro, 2016*; *Cao et al., 2016*) that maintains response time independent of mean stimulus intensity (*Martelli et al., 2013*; *Gorur-Shandilya et al., 2017*), along with a diversity of temporal firing patterns in response to a panel of monomolecular odorants (*Hallem and Carlson, 2006*; *Montague et al., 2011*; *Brown et al., 2005*; *Gupta and Stopfer, 2014*; *Raman et al., 2010*) (*Figure 1D–1E*).

## Front-end Weber-Fechner adaptation preserves odor coding among background and intensity confounds

The identity of an odor is encoded by the pattern of ORN firing responses. However, when a novel foreground odor is presented atop an existing background odor, this pattern may depend also on the background odor, rendering ORN responses less informative about foreground odor identity. To understand how front-end Weber-Fechner adaptation might help encode novel foreground odors in the presence of background odors, we considered environments containing various combinations of foreground odors s and background odors $\bar{s}$, and asked how similar are the ORN responses r to a given s but different $\bar{s}$.

Since it is not possible to visualize the 50-dimensional space of ORN responses, we projected ORN responses onto a two-dimensional space using t-distributed stochastic neighbor embedding (t-SNE) (*van der Maaten and Hinton, 2008*). Like principle component analysis (PCA), t-SNE allows a visualization of high-dimensional objects in such a way that desirable features of the original dataset are preserved (*Figure 2A*). PCA, for example, retains much of the data variance. t-SNE retains the proximity of an object to its nearest neighbors. Specifically, it constructs a probability distribution $Q_{\mathrm{H}}$ based on pairwise distances between nearby objects, assigning higher probability to closer objects. It then determines where the objects would live in a lower dimensional space, such that the analogous distribution $Q_{\mathrm{L}}$ in this space is most similar to $Q_{\mathrm{H}}$. t-SNE is widely used to cluster objects (in our case, ORN responses r to different foreground odors on top of diverse background odors) by similarity (here, foreground odor identity). However, because t-SNE uses local information from only nearest neighbors, global distances and scales are not preserved (*Zhou and Sharpee, 2018*). Thus, we use t-SNE only for visualization. To more rigorously quantify how foreground identity is preserved in ORN activity, we calculate the mutual information (MI) between foreground odor s and ORN firing rates r in the 50-dimensional space (Materials and methods). The MI quantifies how much information a response contains about the stimulus. High MI means that responses exhibit larger variability for different stimuli than for repeated presentations of the same stimulus. In our case, this would be true if r were uniquely defined for different foregrounds s, irrespective of the background $\bar{s}$. Conversely, the MI would be low if responses varied more by background $\bar{s}$ than by foreground s.

We first examined how an adaptive or non-adaptive ORN repertoire encodes odor identity in an odor environment that contains a foreground odor s atop a background odor $\bar{s}$ (*Figure 2B*). Both odors are sparse mixtures, with $K \ll N$ odorants of similar concentrations, odor 'identity' being the particular set of odorants in the mixture. In the adaptive case, we assume that the system has fully adapted to the background $\bar{s}$ before the foreground s is presented. This is enacted by calculating the firing response to the foreground odor $\mathbf{r}(\mathbf{s})$ only after having set the $\epsilon_a$ in *Equation 2* to their steady state values in response to the background odor $\bar{s}$:

$$\epsilon_a(\bar{\mathbf{s}}) = \ln\left[\frac{1 - A_{0a}}{A_{0a}}\right] - (1 - \beta_a)\ln\left(\frac{1 + \mathbf{K}_a \cdot \bar{\mathbf{s}}}{1 + \mathbf{K}_a^* \cdot \bar{\mathbf{s}}}\right), \tag{5}$$

where we have introduced the new parameter $\beta_a$ to allow us to control the scaling of gain adaptation: for $\beta_a = 0$ the system exactly follows Weber-Fechner's law, while for $\beta_a = 1$ there is no adaptation. For small but nonzero $\beta_a$, the inverse gain scales sub-linearly (see Materials and methods), and

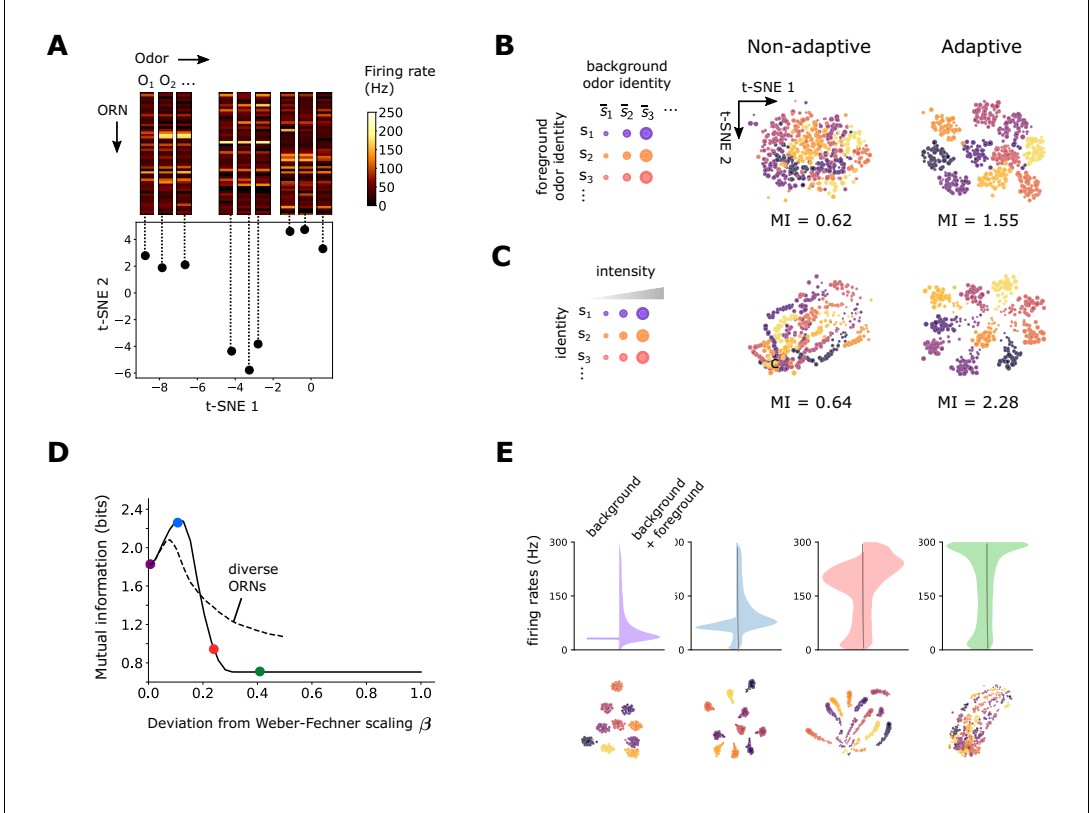

**Figure 2.** Front-end adaptation maintains representations of odor identity across background and intensity confounds. (A) Example t-SNE projection of the 50-dimensional vector of ORN firing rates to two dimensions. Each point represents the firing response to a distinct odor. Nearby points exhibit similarities in corresponding firing rates. (B) t-SNE projection of ORN firing rates, where each point represents the response to foreground odor $\mathbf{s}$ (point color) on top of a background odor $\bar{\mathbf{s}}$ (point size). In the adaptive system, $\epsilon_a$ are set to their steady state values given the background odor $\bar{\mathbf{s}}$ alone according to *Equation 5* with $\beta = 0$. We assumed $A_{0a} = A_0$ for all $a$ (we obtain similar results when $A_{0a}$ are randomly distributed; *Figure 3—figure supplement 1*). Clustering by color implies that responses cluster by foreground odor identity. Since global distances are not preserved by t-SNE, distances between plots cannot be meaningfully compared, and so we do not label the axes with units. Mutual information, in bits, is indicated below the plots. (C) Similar to (B), but now for odors whose concentrations span four decades (represented by point size). Here, the background odor identity is the same for all concentrations. (D) Performance of odor coding as a function of $\beta$, the magnitude of the deviation from Weber-Fechner's law ($\beta = 0$: Weber-Fechner's scaling; $\beta = 1$: no adaptation; see *Equation 5*). Performance is quantified by the mutual information between foreground odor and ORN responses in bits (Materials and methods). Line: same scaling $\beta_a = \beta$ for all ORNs. Dashed: $\beta_a$ is uniformly distributed between 0 and $2\beta<1$ (i.e. has mean $\beta$). (E) Distribution of ORN responses and t-SNE projections for $\beta = 0, 0.10, 0.22, 0.40$ in (D).
DOI: https://doi.org/10.7554/eLife.45293.003

The following figure supplements are available for figure 2:

**Figure supplement 1.** t-SNE projections when background adapted activity level $A_{0a}$ depends on ORN.
DOI: https://doi.org/10.7554/eLife.45293.004

**Figure supplement 2.** Front-end adaptive feedback preserves information capacity of the ORN sensing repertoire.
DOI: https://doi.org/10.7554/eLife.45293.005

the adapted activity $A_a(\bar{\mathbf{s}})$ increases weakly with background $\bar{\mathbf{s}}$. In experiments, small deviations from the strict Weber-Fechner scaling on the order of $\beta \simeq 0.1$ are observed (see extended figures in *Gorur-Shandilya et al., 2017*).

With Weber-Fechner's law in place for all ORNs ($\beta_a = 0$) responses cluster by the identity of foreground odor, showing that the repertoire of ORNs appropriately encodes the identity of novel odors irrespective of background signals – once these backgrounds have been 'adapted away' (*Figure 2B*). This is the case regardless of whether $A_{0a}$ is identical or different across neurons (*Figure 2—figure supplement 1*). In contrast, when the system is non-adaptive, ($\beta_a = 1$), the responses exhibit weaker separations by odor identity (*Figure 2B*). Similarly, responses across different odor intensities are well separated by odor identity in the adaptive system, but less so in the non-adaptive

system (*Figure 2C*). Calculating the mutual information between odor and ORN response in time shows that the adaptive system retains coding capacity as it confronts novel odors (*Figure 2—figure supplement 2*), whereas the non-adaptive system maintains coding capacity in a far more limited range of odor concentration.

To what extent do the benefits of front-end adaptation for odor coding depend on the precise Weber-Fechner scaling? We repeated the analysis from *Figure 2B* for increasing values of $\beta_a = \beta$ between zero (Weber's law) (perfect adaptation) and one (no adaptation). To generalize *Figure 2B*, we now let the intensities range over two decades. As $\beta$ increases, the capacity of the system to cluster responses by odor identity degrades (*Figure 2D*). Introducing diversity among ORNs by distributing $\beta_a$'s uniformly between 0 and $2\beta$ (so that the mean is $\beta$) slightly increases performance at high $\beta$ but reduces it at low $\beta$ (*Figure 2D*). Overall, performance of odor coding degrades with $\beta$, as poorly adapting ORNs begin to saturate (*Figure 2D*).

Interestingly, besides this general trend, we find that for $\beta$ very close to zero, a small deviation from Weber-Fechner's law instead *improves* odor coding. This arises because of the nonlinearity in the onset of adaptation: adaptation kicks in only when the strength of stimulus is sufficient for the response $A_a$ to exceed $A_{0a}$, so that the right hand-side of *Equation 3* is positive. The minimum background intensity $\bar{s}$ required for this to happen is given by $\epsilon_{L,a} = \epsilon_a(\bar{s})$, which, according to equation *Equation 5*, increases with $\beta$. This initial effect increases odor coding performance, as the firing rates can distribute more broadly across the dynamical range of the ORNs, before adaptation is effected (*Figure 2E*). Note that this effect is not specific to our model. A similar enhancement would be observed if Weber's Law were maintained, but kicked in only above a minimum signal intensity. Thus, while Weber-Fechner scaling largely preserves the representation of foreground odor identity amid backgrounds, in some cases it may benefit from a slight relaxation so that the full dynamical range of the ORNs can be exploited.

## Front-end adaptation enhances odor decoding in complex environments

Given that front-end adaptation helps maintain combinatorial odor codes in the presence of backgrounds, we wondered how it affects the capability to decode odor signals from ORN response. One potentially complicating factor is the disparity between sensor dimension and stimulus dimension: while *Drosophila* only express ~60 Or genes (*Vosshall et al., 2000*), the space of odorants is far greater (*Krishnamurthy et al., 2017*). An $N$-dimensional odor signal would naively need $N$ sensory neurons to decode it – one for each odorant. However, naturally occurring odors are sparse, typically comprised of only a few odorants. Enforcing sparsity of the signal during decoding greatly restricts the number of possible odors consistent with a given ORN response, suggesting that such high-dimensional signals might be inferred from less than $N$ ORNs. Indeed, the decoding of sufficiently sparse signals from lower dimensional responses is rigorously guaranteed by the theory of compressed sensing (CS) (*Donoho, 2006*; *Candès et al., 2006*). It is unknown whether CS is implemented in the *Drosophila* olfactory circuit (*Pehlevan et al., 2017*). Here, we use this framework mainly as a tool to quantify how front-end adaptation potentially affects odor decoding, later verifying our conclusions with other classification techniques that incorporate the known architecture of the olfactory system.

CS is performed as a constrained linear optimization. The constraints in the optimization are $\mathbf{r} = \mathbf{D}\mathbf{s}$, where $\mathbf{s}$ is the stimulus to be estimated, $\mathbf{D}$ is the response matrix, and $\mathbf{r}$ is the vector of ORN responses. The cost function to be minimized, $C = \sum_i |s_i|$, enforces sparsity by driving the estimate of each odorant component to zero; the constraints balance this tendency by simultaneously enforcing information from the ORN firing rates. The result is a reconstructed odor signal $\hat{\mathbf{s}}$ that is as sparse as possible, consistent with the ORN responses. In practice, one uses a linear optimization routine to numerically minimize $\sum_i |s_i|$ over $s_i$, subject to $\mathbf{r} = \mathbf{D}\mathbf{s}$. The result is an estimate of the magnitude of each signal component $s_i$. Thus, both the identity and the intensity of the odor signal are estimated.

To incorporate this linear framework of CS into our nonlinear odor encoding model, we treat the nonlinear odor encoding exactly, but approximate the decoding to first order around the background concentration (*Figure 3A*). Specifically, we use *Equations 2-4* to generate ORN responses $\mathbf{r}$ for sparse odors $\mathbf{s}$ having $K \ll N$ nonzero components $s_i = \bar{s}_i + \Delta s_i$, where the mean concentration is

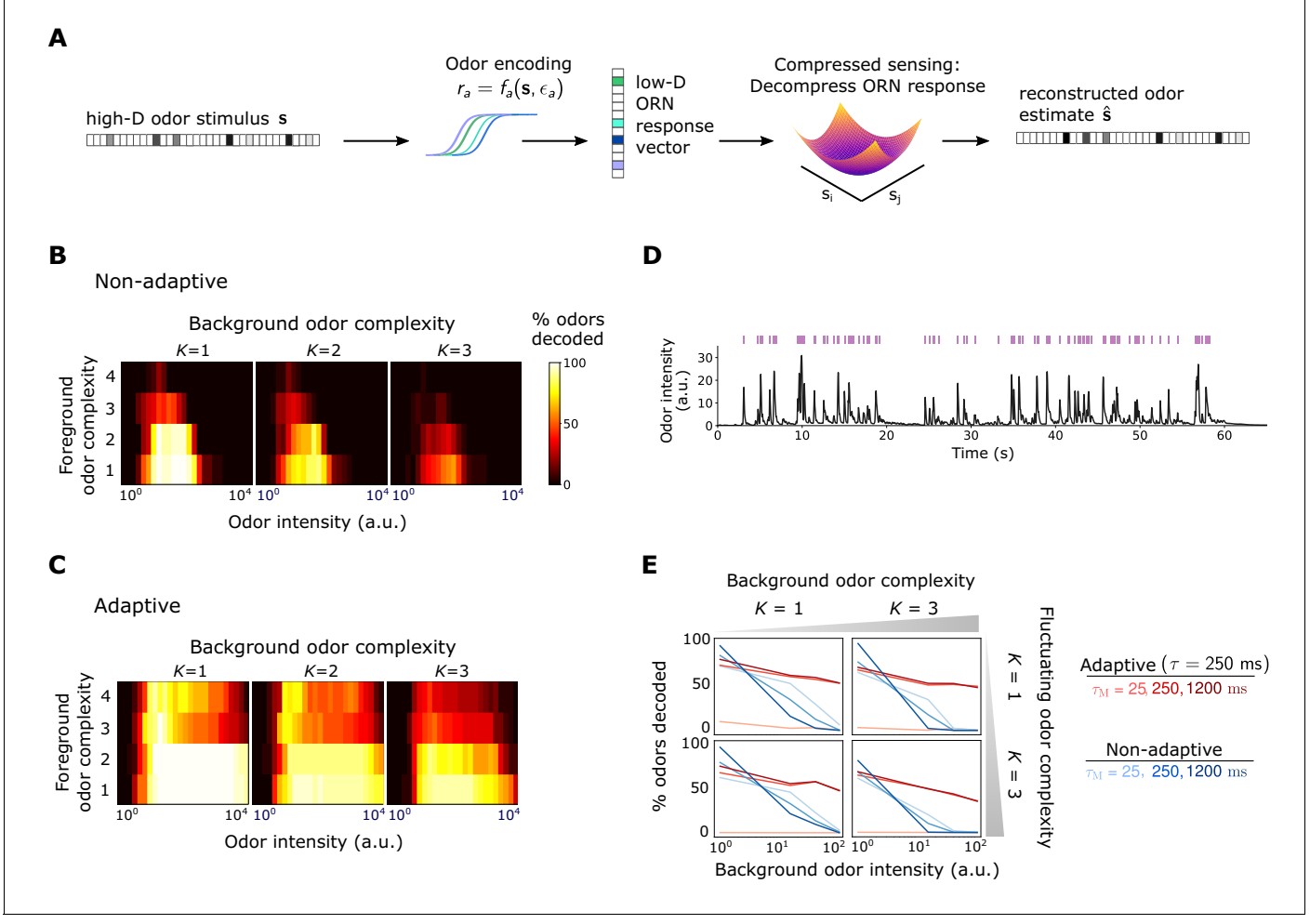

**Figure 3.** Front-end adaptation promotes accurate odor decoding in static and naturalistic odor environments. (**A**) Odor stimuli produce ORN responses via odor-binding and activation and firing machinery, as described by *Equations 2-4*. Odors are then decoded using compressed sensing optimization. Odors are assumed sparse, with $K$ nonzero components, $K \ll N$. (**B**) Decoding accuracy of foreground odors in the presence of background odors, for a system without Weber Law adaptation. (**C**) Same as (**B**), with Weber Law adaptation. (**D**) Recorded trace of naturalistic odor signal; whiffs (signal > 4 a.u.) demarcated by purple bars. This signal is added to static backgrounds of different intensities and complexities. (**E**) Individual plots show the percent of accurately decoded odor whiffs as a function of background odor intensity, for the non-adaptive (blue) and adaptive (red) systems, for different $\tau_M$ (line shades).

DOI: https://doi.org/10.7554/eLife.45293.006

The following figure supplements are available for figure 3:

**Figure supplement 1.** Decoding accuracy for system with ORN-dependent adaptive timescales τ.
DOI: https://doi.org/10.7554/eLife.45293.007

**Figure supplement 2.** Decoding accuracy for system with ORN-dependent adapted firing rates.
DOI: https://doi.org/10.7554/eLife.45293.008

**Figure supplement 3.** Decoding accuracy for receptors with multiple binding sites.
DOI: https://doi.org/10.7554/eLife.45293.009

**Figure supplement 4.** Preservation of restricted isometry property in CS shows how decoding accuracy is maintained by adaptation.
DOI: https://doi.org/10.7554/eLife.45293.010

**Figure supplement 5.** Odor decoding accuracy using the iterative hard thresholding algorithm for nonlinear compressed sensing.
DOI: https://doi.org/10.7554/eLife.45293.011

**Figure supplement 6.** Whiff duration distribution in naturalistic stimulus.
DOI: https://doi.org/10.7554/eLife.45293.012

$\bar{s}_i$. To estimate signals using CS, we minimize $\sum_i |\Delta s_i|$ while enforcing the constraints $\mathbf{r} = \mathbf{D}\Delta\mathbf{s}$, where $\mathbf{D}$ is the linearization of *Equation 2* around $\bar{s}_i$ (details in Materials and methods). The perturbations are chosen as $\Delta s_i \sim \mathcal{N}(s_0/3, s_0/9)$, where $\bar{s}_i = s_0$. This linearization simplifies the CS decoding – namely it enforces a single, global minimum – but it is not critical for our general results; see Materials and methods and *Figure 3—figure supplement 5*. We perform the minimization using the sequential least squares algorithm, producing an estimate of the concentration $\Delta s_i$ of each individual odorant. The matrix $\mathbf{D}$ depends on $\epsilon_a$, and as above, we assume precise adaptation by setting $\epsilon_a$ to their steady state values in response to the background odor alone (via *Equation 5* with $\beta = 0$). In the nonadaptive case, $\epsilon_a$ are held at their minimum values $\epsilon_{L,a}$.

We first examine how foreground odors are recognized when mixed with background odors of a distinct identity but similar intensities, quantifying decoding accuracy as the percentage of odors correctly decoded within some tolerance (see Materials and methods). Without adaptation, accuracy is maintained within the range of receptor sensitivity for monomolecular backgrounds but is virtually eliminated as background complexity rises (*Figure 3B*). The range of sensitivity is broader in the adaptive system and is substantially more robust across odor concentration and complexity (*Figure 3C*).

In realistic odor environments, the concentration and duration of individual odor whiffs vary widely (*Celani et al., 2014*). We wondered how well a front-end adaptation mechanism with a single timescale $\tau$ could promote odor identity detection in such environments. As inputs to our coding/decoding framework, we apply a naturalistic stimulus intensity recorded from a photo-ionization detector (*Gorur-Shandilya et al., 2017*) (*Figure 3D*), to which we randomly assign sparse identities from the $N$-dimensional odorant space (odor concentration fluctuates in time, but identity is fixed). To mimic background confounds, we combine these signals with a static odor background of a different identity. We decode the odor at each point in time using CS optimization. To assess performance, we consider decoding accuracy only during odor whiffs, rather than blanks, where the concentration is too low to be perceived. We assess performance by the percentage of correctly decoded whiffs (signal must be fully decoded at some point during the whiff), and average our results over distinct choices of foreground and background identity. Finally, we assume the decoder has short-term memory: detected odor signals are only retained for $\tau_M$ seconds in the immediate past, bounding the amount of past information utilized in signal reconstruction.

Without ORN adaptation, sufficiently strong backgrounds eliminate the ability to reconstruct the identity of individual odor whiffs, irrespective of the complexity of either the foreground or background odor (*Figure 3E*, blue lines). In the adaptive system, this is substantially mitigated (red lines in *Figure 3E*), provided the memory duration $\tau_M$ is at least as long as the adaptation timescale $\tau$ (darker red lines). The memory $\tau_M$ must be long enough so that information about the background concentration $\bar{s}_i$, which is needed for decoding, can be acquired over a window at least as long as the adaptation timescale. Because short-term adaptation depends on the activity of the Or-Orco channel rather than on the identity of the receptor (*Nagel and Wilson, 2011*; *Martelli et al., 2013*; *Gorur-Shandilya et al., 2017*), the values of $\tau$ and $A_{0a}$ were assumed the same for all ORNs; still, our results hold if these invariances are relaxed (*Figure 3—figure supplement 1* and *Figure 3—figure supplement 2*).

## Front-end adaptation enhances primacy coding

The primacy coding hypothesis has recently emerged as an intriguing framework for combinatorial odor coding. Here, odor identity is encoded by the set (but not temporal order) of the $p$ earliest responding glomeruli/ORN types, known as primacy set of order $p$ (*Wilson et al., 2017*). If the activation order of ORNs were invariant to the strength of an odor step or pulse, primacy sets would in principle form concentration-invariant representation of odor identity. Although our coding framework uses the full ORN ensemble in signal reconstruction, some of these responses may contain redundant information, and a smaller primacy subset may suffice. To examine this, we apply our model to a sigmoidal stimulus that rises to half-max in 50 ms, calculating decoding accuracy in time. Since ORNs activate sequentially, the primacy set is defined by the ORN subset active when the odor is decoded. For simple odors, a limited set of earliest responding neurons fully accounts for the odor identity (*Figure 4A*), in agreement with primacy coding. As expected for more complex odor mixtures, the full repertoire is required for accurate decoding. Primacy coding also predicts

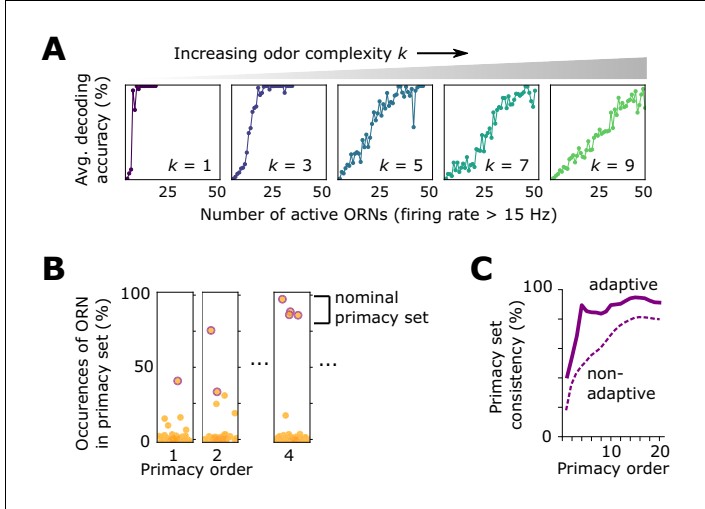

**Figure 4.** Effect of front-end adaptation on primacy coding. (**A**) Decoding accuracy as a function of the number of active ORNs, for different odor complexities. The primacy set consists of those ORNs required to be active for accurate decoding. (**B**) Frequency of particular ORNs in primacy sets of an odor placed atop different backgrounds. Individual plots show, for given primacy order $p$, the percentage of backgrounds for which the primacy set of odor A contains a given ORN (dots). Those with purple borders are the $p$ most highly occurring – that is a nominal background-invariant primacy set for odor A. Points are jittered horizontally for visualization. (**C**) Consistency of primacy sets across backgrounds, as a function of $p$, for the adaptive (solid) and non-adaptive (dashed) system. Consistency is defined as the likelihood that an ORN in the nominal primacy set appears in any of the individual background-dependent primacy sets, averaged over the nominal set (average of the y-values of the purple dots in **B**). 100% consistency means that for all backgrounds, the primacy set of odor (**A**) is always the same $p$ ORNs.

DOI: https://doi.org/10.7554/eLife.45293.013

The following figure supplement is available for figure 4:

**Figure supplement 1.** Additional results for primacy coding in the adaptive ORN model.

DOI: https://doi.org/10.7554/eLife.45293.014

that for stronger stimuli, responses occur earlier, since the primacy set is realized quicker, which our framework replicates (*Figure 4—figure supplement 1*).

Beyond mere consistency, however, front-end adaptation might also enhance primacy coding in different environments, such as background odors, which could scramble primacy sets. To investigate this, we considered again a sigmoidal odor step (odor A), now atop a static background (odor B) to which the system has adapted. We compared the primacy sets of odor A for 1000 different choices of odor B, finding that, with adaptation, primacy sets are highly consistent across background confounds for all but the smallest primacy orders (*Figure 4B*-Figure 4C). This also holds true for backgrounds of different concentrations (*Figure 4—figure supplement 1*), suggesting a central role for front-end adaptation in reinforcing primacy codes across differing environmental conditions.

## Contribution of front-end adaptation for odor recognition within the *Drosophila* olfactory circuit

Signal transformations in the sensing periphery are propagated through the remainder of the olfactory circuit. How does front-end adaptation interact with these subsequent neural transformations? ORNs expressing the same OR converge to a unique AL glomerulus, where they receive lateral inhibition from other glomeruli (*Olsen and Wilson, 2008*; *Asahina et al., 2009*). This inhibition implements a type of divisive gain control (*Olsen et al., 2010*), normalizing the activity of output projections neurons, which then synapse onto a large number of Kenyon cells (KCs) in the mushroom body. To investigate how odor representations are affected by interactions between front-end ORN adaptation and this lateral inhibition and synaptic divergence, we extended our ORN encoding model by adding uniglomerular connections from ORNs to the antennal lobe, followed by sparse,

divergent connections to 2500 KCs (*Keene and Waddell, 2007*; *Litwin-Kumar et al., 2017*; *Caron et al., 2013*). Inhibition was modeled via divisive normalization, with parameters chosen according to experiment (*Olsen et al., 2010*). We quantified decoding accuracy by training and testing a linear classifier on the KC activity output of sparse odors of distinct intensity and identity. We trained the classifier on $N_{ID}$ sparse odor identities at intensities chosen randomly over 4 orders of magnitude, then tested the classifier accuracy on the same set of odor identities but of differing concentrations.

With both ORN adaptation and divisive normalization, the accuracy of the classification by odor identity remains above 85% for more than 1000 odor identities ($N_{ID}>1000$). Removing ORN adaption while maintaining divisive normalization substantially reduces accuracy (down to 65% for 1000 odor identities). Further removing divisive normalization gives similar results, apart for very large numbers of odors identities ($N_{ID}>1000$), where divisive normalization provides benefits (*Figure 5A*). These results strongly implicate front-end adaptation as a key player in maintaining odor identity representations, before signals are further processed downstream.

As a simpler task, we also considered binary classification, categorizing odors as appetitive or aversive. For simplicity, odor signals of the same identity but differing intensity were assigned the same valence. Classification accuracy degrades to chance level as $N_{ID}$ becomes very large (*Figure 5B*). When acting alone, either divisive normalization or ORN adaptation can help, although the effect of ORN adaptation is slightly stronger. When both are active, accuracy improves further, suggesting that these distinct adaptive transformations may act jointly at different stages of neural processing in preserving representations of odor identity. As expected, these gains mostly vanish for the same odors chosen from a narrower range of concentrations (*Figure 5—figure supplement 1*).

Previous simulation results have shown that divisive normalization aids identity decoding from PN response to a stronger degree than we find here (*Olsen et al., 2010*). There, 19 distinct odor identities at three concentrations were classified more accurately with divisive normalization (80%) than without (68%). In our case, we find about ~75% accuracy, with and without divisive normalization. This discrepancy is not necessarily inconsistent. First, we decode mixtures, not single odorants, and the combinatorics may reduce the benefit of divisive normalization. Second, we classify the responses of 2500 KCs, rather than 50 PNs (or 24 PNs as in *Olsen et al., 2010*). Kenyon cell responses follow a high degree of postsynaptic divergence from PNs, which could decorrelate neural responses (*Caron et al., 2013*; *Litwin-Kumar et al., 2017*; *Krishnamurthy et al., 2017*) similarly to divisive normalization, reducing the gains from the latter. Finally, the divisive normalization model is a simple one in which glomeruli are all mutually inhibiting. A more complex model in which each

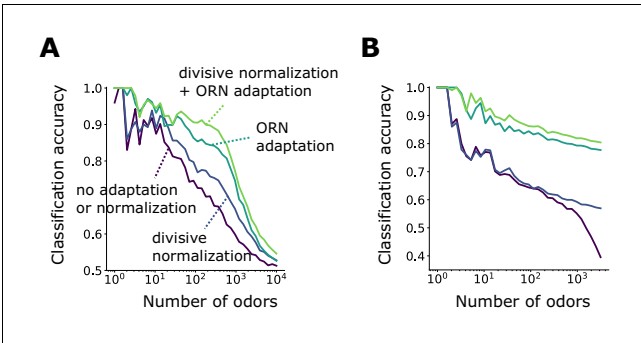

**Figure 5.** Front-end adaptation enhances odor recognition by the *Drosophila* olfactory circuit. (**A**) Accuracy of linear classification by odor identity, as a function of the number of distinct odor identities classified by the trained network (concentrations span 4 orders of magnitude), in systems with only ORN adaptation, only divisive normalization, both or neither. (**B**) Same as (**A**) but now classifying odors by valence. Odors were randomly assigned valence. For a given odor identity, the valence is the same for all concentrations.
DOI: https://doi.org/10.7554/eLife.45293.015

The following figure supplement is available for figure 5:

**Figure supplement 1.** Binary classification for odors whose concentrations span a narrow range of concentration.
DOI: https://doi.org/10.7554/eLife.45293.016

glomerulus inhibits only a subset of other glomeruli through local neurons might produce a larger contribution.

In sum, these results indicate that ORN adaptation might contribute significantly to odor recognition by identity and valence, and that divisive normalization also contributes, although possibly more to classification by valence than by identity. An intriguing possibility is that these two forms of gain control play different roles in coding discrete odor categories versus odor identities.

## Discussion

Weber-Law adaptation at the very front-end of the insect olfactory circuit (*Gorur-Shandilya et al., 2017*; *Cafaro, 2016*; *Cao et al., 2016*) may contribute significantly to the preservation of neural representations of odor identity amid confounding odors and intensity fluctuations. Drawing on experimental evidence for a number of ORN-invariant response features (*Nagel and Wilson, 2011*; *Martelli et al., 2013*; *Stevens, 2016*; *Gorur-Shandilya et al., 2017*; *Si et al., 2019*), we have found that this mechanism of dynamic adaptation confers significant benefits in coding fidelity, without the need for ORN-specific parameterizations. Still, our results hold when these invariances such as adaptation timescale or baseline activity are relaxed (*Figure 3—figure supplement 1* and *Figure 3—figure supplement 2*). In the olfactory periphery, front-end Weber Law adaptation therefore appears fairly robust, a consequence of controlling gain via feedback from channel activity (*Waite et al., 2018*; *Nagel and Wilson, 2011*; *Gorur-Shandilya et al., 2017*), rather than through intrinsic, receptor-dependent mechanisms.

Our results also suggest that a slight breaking of Weber scaling may aid combinatorial coding, by spreading firing rates more fully over the ORN dynamic range, while still preventing saturation. The degree of this breaking would manifest as a correction to the Weber scaling exponent, $\sim (1/s)^1 \rightarrow \sim (1/s)^{1-\beta}$, which could in principle be measured experimentally for individual ORNs. Such small deviations from the strict Weber-Fechner scaling have been observed (see extended figures in *Gorur-Shandilya et al., 2017*).

While our framework incorporates many observed features of the *Drosphila* olfactory system – Weber-Law adaptation, power-law distributed receptor affinities, temporal filter invariance, connectivity topologies – it is minimal. We considered only one of the chemoreceptor families expressed in the fly antenna (*Joseph and Carlson, 2015*) and ignored possible contributions of odor binding proteins (*Vogt and Riddiford, 1981*; *Menuz et al., 2014*), inhibitory odorants (*Cao et al., 2017*), and odorant-odorant antagonism (*Reddy et al., 2018*), which could further boost coding capacity and preserve representation sparsity. Useful extensions to our nonlinear-linear-nonlinear model might incorporate ephaptic coupling between ORNs housed in the same sensillum (*Su et al., 2012*), global inhibition in the mushroom body (*Papadopoulou et al., 2011*), and the effects of long-term adaptation (*Guo et al., 2017*).

Previous studies have characterized various neural mechanisms that help preserve combinatorial codes. Lateral inhibition between glomeruli helps tame saturation and boost weak signals (*Olsen et al., 2010*). The sparse degree of connectivity to either the olfactory bulb (vertebrates) or mushroom body (insects) may also be precisely tuned to optimize the capacity to learn associations (*Litwin-Kumar et al., 2017*). In this work, we find that some of these downstream features act in concert with front-end dynamic adaptation in maintaining representations of odor identity.

Other studies have implicated the unique temporal patterns of neural response as signatures of odor identity (*Raman et al., 2010*; *Gupta and Stopfer, 2011*; *Brown et al., 2005*; *Gupta and Stopfer, 2014*). ORN and projection neuron time traces form distinct trajectories in low-dimensional projections, and cluster by odor identity, much as we have found here for static responses at different concentrations (*Figure 2*). In locusts PNs, the trajectories elicited by foreground odors when presented in distinct backgrounds exhibit some degree of overlap; although partial, these overlaps were nonetheless sufficient to maintain background-invariant decoding from Kenyon cell responses (*Saha et al., 2013*). It was therefore suggested that background filtering likely occurs at the level of ORNs themselves (*Saha et al., 2013*). Likewise, in our framework, temporal coding is implicit: because the input nonlinearity depends on the diversity of binding affinities, odor signals are naturally formatted into temporal patterns that are both odor- and ORN-specific (*Figure 1D-Figure 1E*). Further, the short required memory timescales ($\tau_M \sim \tau \sim 250\,\mathrm{ms}$) suggest that only brief time windows are needed for accurate odor identification, consistent with previous findings (*Brown et al., 2005*;

*Saha et al., 2013*). Moreover, we find that front-end adaptation enhances the robustness of other combinatorial coding schemes, such as primacy coding (*Wilson et al., 2017*), which relies on the temporal order of ORN activation but not absolute firing rate (*Figure 4*).

In the well-characterized chemosensory system of bacterial chemotaxis, Weber Law adaptation is enacted through a feedback loop from the output activity of the receptor-kinase complexes onto the enzymes modifying receptor sensitivity (*Waite et al., 2018*). It is interesting that some aspects of this logic are also present in ORNs: although the molecular players are different (and still largely unknown, although likely involving calcium channel signaling, *Cao et al., 2016*), it has been shown that transduction activity feeds back onto the sensitivity of Or-Orco ligand-gated cation channels, enabling the Weber-Fechner relation (*Nagel and Wilson, 2011*; *Gorur-Shandilya et al., 2017*; *Cao et al., 2016*). That this adaptation mechanism appears to act similarly across ORNs (*Gorur-Shandilya et al., 2017*; *Martelli et al., 2013*; *Cao et al., 2016*) suggests the possible involvement of the universal co-receptor Orco, whose role in long-term adaptation has recently been reported (*Getahun et al., 2013*; *Getahun et al., 2016*; *Guo et al., 2017*). Further, the identification of four subunits comprising the Orco-Or ion channel suggest that generic Or/Orco complexes may contain multiple odorant binding sites, which when included in our model supports our general findings (*Figure 3—figure supplement 3*).

Weber Law ensures that sensory systems remain in the regime of maximum sensitivity, broadening dynamic range and maintaining information capacity (*Wark et al., 2007*). For a single-channel system, this requires matching the midpoint of the dose-response curve to the mean ligand concentration (*Nemenman, 2012*), a strategy which may fail in multi-channel systems with overlapping tuning curves: adaptation to one signal could inhibit identification of others, if the signals excite some but not all the same sensors, as in *Figure 1G*. Our results show that this strategy is still largely functional. In CS decoding, this can be traced to the observation that accuracy is guaranteed when sufficiently distinct odor identities produce sufficiently distinct ORN responses, a condition known as the restricted isometry property (*Candès et al., 2006*). Indeed, the Weber-Fechner scaling increases the likelihood that this property is satisfied, beyond that in the non-adaptive system (SI text and *Figure 3—figure supplement 4 - Figure 3—figure supplement 5*). Still, restricted isometry does not require that response repertoires are *invariant* to environmental changes. That is, even if the subset of active ORNs were concentration-dependent, odors could still in principle be fully reconstructible by CS. Nonetheless, our results in t-SNE clustering (*Figure 2*), primacy coding (*Figure 4B–4C*), and odor classification (*Figure 5*) suggest that some signature of response invariance emerges as a natural byproduct of front-end adaptation. Together, this implies that Weber Law adaptation, whether required by the olfactory circuit for precise signal reconstruction (as in CS) or for developing odor associations (as in classification), can play an integral part in maintaining combinatorial codes amid changing environmental conditions.

## Materials and methods

### Adaptive ORN model

We model an odor as an $N$-dimensional vector $\mathbf{s} = [s_1, ..., s_N]$, where $s_i > 0$ are the concentrations of individual volatile molecules (odorants) comprising the odor. The olfactory sensory system is modeled as a collection of $M$ distinct Or/Orco complexes indexed by the sub index $a = 1, ..., M$, each of which can be bound with any one of the odorant molecules, and can be either active (firing) or inactive (quiescent). At first, we assume there is one binding site per complex; this will be generalized to many sites. We consider the binding and activation processes to be in equilibrium, assigning each state a corresponding Boltzmann weight, where the zero of energy is set by the unbound, inactive state $C_a$. These weights are:

$$
\begin{array}{ll}
\mathrm{C_a} & 1 \\
\mathrm{C_a^*} & \exp(-\beta \epsilon_a) \\
\mathrm{C_a : s_i} & \exp(-\beta(-E_{ai} - \mu_i)) \\
\mathrm{C_a^* : s_i} & \exp(-\beta(-(E_{ai}^* - \epsilon_a) - \mu_i)),
\end{array} \tag{6}
$$

where $\epsilon_a$ (assumed positive) is the free energy difference between the active and inactive

conformation of the unbound receptor, and $E_{ai}$ and $E_{ai}^*$ are the free energy differences (assumed positive) between the unbound and bound state for the inactive and active receptor, respectively. $\mu_i = \mu_0 + \beta^{-1}\log(s_i/s_0)$ is the chemical potential for odorant species $i$ in terms of a reference chemical potential $\mu_0$ at concentration $s_0$, $s_0 \exp(-\beta\mu_0) = s_i \exp(-\beta\mu_i)$, which can be traded for the thermodynamic-relevant disassociation constants $K_{ai}^{-1} = K_{D,ai} = s_0 e^{\beta(-E_{ai}-\mu_0)}$.

Adding up contributions from all $i$ odorants, the active fraction is:

$$
\begin{aligned}
A_a &= \frac{C_a^* + \sum_i C_a^*:s_i}{C_a^* + \sum_i C_a^*:s_i + C_a + \sum_i C_a:s_i} \\
&= \left(1 + \frac{C_a + \sum_i C_a:s_i}{C_a^* + \sum_i C_a^*:s_i}\right) \\
&= \left(1 + e^{\epsilon_a}\frac{1 + \mathbf{K_a}\cdot\mathbf{s}(t)}{1 + \mathbf{K_a^*}\cdot\mathbf{s}(t)}\right)^{-1},
\end{aligned}
\tag{2}
$$

where we have expressed free energies in units of $k_B T = \beta^{-1}$ for notational convenience.

This expression can be generalized for the case of multiple, independent binding sites through some simple combinatorial factors. Consider first an odorant $i$ which can bind one of two locations on receptor $a$. There are then four possible inactive states: both sites unbound, site one bound, site two bound, both sites bound. Combined with the active states, there are therefore eight states for odorant $i$ and receptor $a$, with energies:

$$
\begin{aligned}
\text{active} \quad & \{1, \; -E_{ai}-\mu_i, -E_{ai}-\mu_i, -2E_{ai}-2\mu_i\} \\
\text{inactive} \quad & \{\epsilon_a, -(E_{ai}^*-\epsilon_a)-\mu_i, -(E_{ai}^*-\epsilon_a)-\mu_i, -(2E_{ai}^*-\epsilon_a)-2\mu_i\}
\end{aligned}
\tag{7}
$$

In the active fraction, *Equation 2*, the Boltzmann factors combine through the binomial theorem, giving (for a single odorant environment $i$):

$$
\begin{aligned}
&A_a(\text{odorant i}, 2\,\text{binding sites}) \\
&= \left[1 + e^{\epsilon_a}\left(\frac{1 + \mathbf{K}_a\cdot\mathbf{s}(t)}{1 + \mathbf{K}_a^*\cdot\mathbf{s}(t)}\right)^2\right]^{-1}.
\end{aligned}
\tag{8}
$$

This expression generalizes for an arbitrary number of odorants and independent binding sites through the appropriate combinatorial factors, giving an active fraction of

$$
\begin{aligned}
&A_a(N\,\text{odorants}, R\,\text{binding sites}) \\
&= \left[1 + e^{\epsilon_a}\left(\frac{1 + \mathbf{K}_a\cdot\mathbf{s}(t)}{1 + \mathbf{K}_a^*\cdot\mathbf{s}(t)}\right)^R\right]^{-1}.
\end{aligned}
\tag{9}
$$

To generate ORN time traces, *Equations 2-3* are integrated numerically using the Euler method with a 2 ms time step. For ORN firing (*Equation 4*), $h(t)$ is bi-lobed (*Martelli et al., 2013*): $h(t) = A p_{\text{Gam}}(t; \alpha_1, \tau_1) - B p_{\text{Gam}}(t; \alpha_2, \tau_2)$, $A = 190$, $B = 1.33$, $\alpha_1 = 2$, $\alpha_2 = 3$, $\tau_1 = 0.012$, and $\tau_2 = 0.016$, where $p_{\text{Gam}}$ is the pdf of Gamma($\alpha$, $1/\tau$). Nonlinearity $f$ is modeled as a linear rectifier with 5 Hz threshold.

## Derivation of ORN gain

Weber's Law states that the gain, or differential response, of the receptor activity $A_a$ scales with the mean odor concentration $\bar{s}_i$. To show how this is satisfied in our model, we consider the response, *Equation 2*, to a signal $\mathbf{s} = \bar{\mathbf{s}} + \Delta\mathbf{s}$, where $\Delta\mathbf{s}$ consists of only a small fluctuation in the $i$th component $\Delta s_i < |\bar{s}_i|$ about the mean. We derive the change in response to fluctuation $\Delta s_i$ for general $\beta$ from 0 (Weber's law) to 1 (no adaptation).

First we write the activity in the form:

$$
A_a = (1 + e^{F_a})^{-1},
\tag{10}
$$

where

$$
F_a = \epsilon_a(\bar{\mathbf{s}}) + \ln\left(\frac{1 + \mathbf{K}_a\cdot\mathbf{s}}{1 + \mathbf{K}_a^*\cdot\mathbf{s}}\right),
\tag{11}
$$

where $\epsilon_a(\bar{\mathbf{s}})$ is given by **Equation 5**. Then, assuming $1/\mathbf{K}_a^* \ll s_i \ll 1/\mathbf{K}_a$, the change in response from the adapted level $A_a(\bar{\mathbf{s}})$ is

$$A_a(\mathbf{s}) - A_a(\bar{\mathbf{s}}) = \Delta A_a \quad = \frac{dA_a}{dF_a}\frac{dF_a}{ds}\big|_{\bar{\mathbf{s}}}\Delta s_i$$

$$= -\frac{e^{F_a}}{(1+e^{F_a})^2}\big|_{\bar{\mathbf{s}}}\left(\frac{-K_{ai}^*}{\mathbf{K}_a^*\cdot\bar{\mathbf{s}}}\right)\Delta s_i. \tag{12}$$

We use **Equation 5** to evaluate $e^{F_a}$ at $\bar{\mathbf{s}}$, obtaining:

$$e^{F_a} \approx \frac{1-A_{0a}}{A_{0a}}(\mathbf{K}_a^*\cdot\bar{\mathbf{s}})^{-\beta}, \tag{13}$$

whereby

$$\frac{\Delta A_a}{\Delta s_i} \quad = \frac{\frac{1-A_{0a}}{A_{0a}}(\mathbf{K}_a^*\cdot\bar{\mathbf{s}})^{-\beta}}{(1+\frac{1-A_{0a}}{A_{0a}}(\mathbf{K}_a^*\cdot\bar{\mathbf{s}})^{-\beta})^2}\left(\frac{K_{ai}^*}{\mathbf{K}_a^*\cdot\bar{\mathbf{s}}}\right)$$

$$= \frac{(1-A_{0a})A_{0a}K_{ai}^*}{[A_{0a}(\mathbf{K}_a^*\cdot\bar{\mathbf{s}})^{\frac{1+\beta}{2}}+(1-A_{0a})(\mathbf{K}_a^*\cdot\bar{\mathbf{s}})^{\frac{1-\beta}{2}}]^2}. \tag{14}$$

For $\beta = 0$ (the fully adaptive case) and a single odorant, this expression for the gain reduces to $(1-A_{0a})A_{0a}/s_i$. For small $\beta$, and given $A_{0a} \simeq 0.1$ (corresponding to 30 Hz on a 300 Hz firing rate scale), the denominator is dominated by the $1-A_{0a}$ term, giving:

$$\frac{\Delta A_a}{\Delta s_i}\big|_{(\beta\ll1)} = \frac{A_{0a}K_{ai}^*}{(1-A_{0a})(\mathbf{K}_a^*\cdot\bar{\mathbf{s}})^{1-\beta}}. \tag{15}$$

The implication of this is that the gain scaling of the inverse mean intensity, which is 1 for perfect adaptation (gain $\sim(1/s_i)^1$), is now sublinear. Thus, when Weber's Law is weakly broken, the gain still reduces with mean odor intensity, but not as quickly.

## t-SNE dimensionality reduction and mutual information

For t-SNE dimensionality reduction (**van der Maaten and Hinton, 2008**), ORN responses were generated for odor signal combinations consisting of 1 (among 10) distinct sparse foreground odors A atop 1 (among 50) distinct sparse background odors B, for **Figure 2B**. **Figure 2C** plots responses for 10 odors at 40 concentrations spanning four decades, atop a random sparse background odor of similar magnitude. For adaptive systems, $\epsilon_a$ were set to their fully adapted values to the background odor, given by **Equation 5**, with $\beta = 0$.

The mutual information (MI) between signal and response quantifies how many bits of information a response contains about the stimulus. As we are interested in how much information ORN responses $\mathbf{r}$ contain about novel foreground odors $\mathbf{s}$, we calculate the MI between $\mathbf{s}$ and $\mathbf{r}$. This calculation requires the conditional response distribution $P(\mathbf{r}|\mathbf{s})$, where the probability distribution is over different background odors $\bar{\mathbf{s}}$. To get this distribution, we hold $\mathbf{s}$ fixed and generate $\mathbf{r}$ in the presence of distinct backgrounds. To these responses $\mathbf{r}$, we also add a small amount of Gaussian noise (mean zero and variance 1 Hz), which allows a distribution to be defined when there is no background odor. We then bin the resulting $\mathbf{r}$ in units of $\Delta r = 1$ Hz to get a histogram representing $P(\mathbf{r}|\mathbf{s})$ (The histogram is necessary, since a sum must be taken over $\mathbf{r}$). If responses were completely background invariant, the resulting histogram would be highly peaked.

Using $P(\mathbf{r}|\mathbf{s})$, the MI is defined as

$$\mathrm{MI} = H_{\mathrm{response}} - H_{\mathrm{noise}}$$

where $H_{\mathrm{noise}}$ is:

$$H_{\mathrm{noise}} = -\sum_{\mathbf{s},\mathbf{r}}P(\mathbf{s})P(\mathbf{r}|\mathbf{s})\log_2 P(\mathbf{r}|\mathbf{s})$$

and $H_{\mathrm{response}}$ is

$$H_{\text{response}} = -\sum_{\mathbf{r}} P(\mathbf{r}) \log_2 P(\mathbf{r})$$

where

$$P(\mathbf{r}) = \sum_{\mathbf{s}} P(\mathbf{s})P(\mathbf{r}|\mathbf{s}).$$

The noise entropy $H_{\text{noise}}$ quantifies how much variability comes from the background odors, but is not related to changes in foreground odor. The response entropy $H_{\text{response}}$ quantifies how much variability comes from both background and foreground. The mutual information, which is their difference, is a measure of how responses differ by foreground alone.

## Compressed sensing decoding of ORN responses

Compressed sensing (CS) addresses the problem of determining a sparse signal from a set of linear measurements, when the number of measurements is less than the signal dimension. Specifically, it is a solution to

$$\mathbf{y} = \mathbf{D}\mathbf{x}, \tag{16}$$

where $\mathbf{x} \in \mathbb{R}^N$ and $\mathbf{y} \in \mathbb{R}^M$ are vectors of signals and responses, respectively, and $\mathbf{D}$ is the measurement matrix. Since measurements are fewer than signal components, then $M<N$, whereby $\mathbf{D}$ is wide rectangular and so *Equation 16* cannot be simply inverted to produce $\mathbf{x}$. The idea of CS is to utilize the knowledge that $\mathbf{x}$ is sparse, that is only $K$ of its components, $K \ll N$ are nonzero. Both the measurements and sparsity are thus combined into a single constrained optimization routine:

$$\hat{x}_i = \operatorname{argmin} \sum_i^N |x_i| \quad \text{such that } \mathbf{y} = \mathbf{D}\mathbf{s} \tag{17}$$

where $\hat{x}_i$ are the optimal estimates of the signal components and the sum, which is known as the $L_1$ norm of $\mathbf{x}$, is a natural metric of sparsity (*Donoho, 2006*).

The $L_1$ norm is a convex operation and the constraints are linear, so the optimization has a unique global minimum. To incorporate the nonlinear response of our encoding model into this linear framework, we assume that the responses are generated through the full nonlinear steady state response, *Equations 2- 4*, but that the measurement matrix $\mathbf{D}$ needed for decoding uses a linear approximation of this transformation. Expanding *Equation 4* around $\bar{s} = s - \Delta s$ gives

$$\begin{aligned} \Delta r_a(t) &= r_a(\mathbf{s}(t)) - r_a(\bar{\mathbf{s}}(t)) \\ \Delta r_a(t) &= \int^t d\tau h(t-\tau) \sum_i^N \frac{dA_{ai}}{ds_i}\big|_{\bar{s}} \Delta s_i \end{aligned} \tag{18}$$

where

$$r_a(\mathbf{s}_0) = \int^t d\tau h(t-\tau) \sum_i^N A_{0a} \tag{19}$$

and where $\frac{dA_{ai}}{ds_i}\big|_{\bar{s}}$ is given by the right-hand side of *Equation 14* with $\beta = 0$. *Equations 18 and 19* hold only for integrands above 5 Hz (and are zero below), as per the linear rectifier $f$. We assume that the neural decoder has access to background $\bar{s}$, presumed learned (this assumption can be relaxed; see below), and to the linearized response matrix, *Equation 14*, but must infer the excess signals $\Delta s_i$ from excess ORN firing rates $\Delta r_a(t)$. Thus, this corresponds to the CS framework (*Equation 17*) via $\Delta \mathbf{r} \to \mathbf{y}$, $\Delta \mathbf{s} \to \mathbf{x}$, and $dA_{ai}/ds_i\big|_{\bar{s}} \to \mathbf{D}$. We optimize the cost function in *Equation 17* using sequential least squares programming, implemented in Python through using the scientific package SciPy.

For our simulations, we let sparse components $s_i$ be chosen as $s_i = \bar{s}_i + \Delta s_i$, where $\bar{s}_i = s_0$ and $\Delta s_i \sim \mathcal{N}(s_0/3, s_0/9)$. The measurement matrix $\mathbf{D}$ depends on the free energy differences $\epsilon_a$. For static stimuli, $\epsilon_a$ equals the fixed point of *Equation 3* in response to the background stimulus with $\beta = 0$. For fluctuating stimuli, $\epsilon_a$ is updated in time by continuously integrating $r_a(t)$, via *Equation 3 and 4*; thus, only knowledge of the response $r_a(t)$ are needed by the decoder. To quantify decoding

accuracy, we treat the zero and nonzero components of the sparse odor vector separately. We demand that the $K$ nonzero components $\hat{s}_i$ of the estimated sparse vector are within 25% of their true values $s_i$, and that the $N - K$ zero components are estimated less than 10% of $s_0$. Together, this ensures that the odorants comprising the odor mixture are estimated sufficiently close to their concentrations, and that the remaining components are sufficiently small. Odor signals s are considered correctly decoded if both of these conditions are satisfied for all components $s_i$. The relatively lax accuracy demanded on the nonzero components is to prevent oversensitivity on the unavoidable errors introduced by linearization. Qualitatively, our findings are robust to these choices.

The naturalistic odor signal (*Figure 3D*) was generated by randomly varying flow rates of ethyl acetate and measuring the concentration with a photo-ionization detector (*Gorur-Shandilya et al., 2017*). Statistics mirroring a turbulent flow (*Celani et al., 2014*) were verified (*Figure 3—figure supplement 6*).

## Iterative hard thresholding (IHT) and the restricted isometry property in compressed sensing

The purpose of response linearization (*Equation 18*) is simply to apply compressed sensing reconstruction directly using linear programming, without worrying about issues of local minima in *Equation 17*. This allows us to isolate the impact of Weber Law adaptation from the particularities of the numerics. An alternate technique for compressed signal reconstruction, *iterative hard thresholding* (IHT), does not minimize the constrained $L_1$ norm directly, rather applying a hard threshold to an iteratively updated signal estimate (*Blumensath and Davies, 2009b*). IHT can be generalized straightforwardly to nonlinear constraints, and would actually dispense with the need for a learned background $\bar{s}$, simply initializing the iterations from $\bar{s} = 0$. Remarkably, this technique works quite well even for non-linear measurements (*Blumensath, 2013*). We demonstrate the applicability of the IHT algorithm to our odor decoding system in *Figure 3—figure supplement 5*, which reproduces qualitatively the findings in the main text. For these calculations, no background odor was assumed, each iterative decoding being initialized $\bar{s} = 0$.

IHT provides an alternate computational technique of nonlinear CS, which could be used to both extend and verify our results. Further, it allows us to illustrate why Weber Law adaptation maintains signal reconstruction fidelity in our olfactory sensing model. Like CS using $L_1$-norm minimization, IHT exhibits amenable reconstruction and convergence properties under the guarantee of the so-called restricted isometry property (RIP) (*Candès et al., 2006*). Loosely, RIP measures how closely a matrix operator resembles an orthogonal transformation when acting on sparse vectors. The degree to which RIP is satisfied can be understood in terms of the spectrum of a measurement matrix $\mathbf{D}$. In particular, if $\lambda_i$ are the eigenvalues of $\mathbf{D}_i^T\mathbf{D}_i$, where $\mathbf{D}_i$ is any $k \times m$ submatrix of $\mathbf{D}$, and

$$1 - \delta_i \leq \lambda_{\min} \leq \lambda_{\max} \leq 1 + \delta_i \tag{20}$$

is satisfied for some $\delta_i$, then $\mathbf{D}$ satisfies the RIP with constant $\delta_i$. Plainly, the RIP states that the eigenvalues of $\mathbf{D}_i^T\mathbf{D}_i$, when acting on $k$-sparse vectors, are centered around 1. Thus, to intuit why signal reconstruction breaks down in the non-adaptive sensing system, we can investigate the eigendecomposition of various linearizations of the measurement matrix. We do this now, starting with a brief description of the IHT.

In the linear setting, IHT seeks sparse signals via the following iterative procedure (*Blumensath and Davies, 2009b*):

$$\mathbf{s}_{i+1} = H_K(\mathbf{s}_i + \mu\mathbf{D}^T(\mathbf{s}_i + (\mathbf{y} - \mathbf{D}\mathbf{s}_i))) \tag{21}$$

where $\mathbf{s}_i$ is the $i$th estimate of the sparse signal s, $\mu$ is a step size for the iterations, and $\mathbf{y}$, $\mathbf{D}$ are as defined above. $H_k(\cdot)$ is a thresholding function which sets all but the largest $K$ values of its argument to zero. The nonlinear extension to IHT is (*Blumensath, 2013*):

$$\mathbf{s}_{i+1} = H_K(\mathbf{s}_i + \mu\mathbf{D}_{\mathbf{s}_i}^T(\mathbf{s}_i + (\mathbf{y} - D(\mathbf{s}_i)))), \tag{22}$$

where $D$ is a nonlinear sensing function and $\mathbf{D}_{\mathbf{s}_i}$ is a linearization of $D$ about the point $\mathbf{s}_i$. Reconstructibility for $k$-sparse signals is guaranteed if $\mathbf{D}_{\mathbf{s}_i}$ satisfies RIP for all $\mathbf{s}_i$ and all $k$-sparse vectors (*Blumensath and Davies, 2009b*). To get a sense of how this is preserved in the adaptive system,

we calculate the eigenvalues for 1000 choices of $s_i$, acting on random signals of given sparsity $K$ (*Figure 3—figure supplement 4*). Since the RIP is sensitive to constant scalings of the measurement matrix (while the actual estimation problem is not), we scaled all columns of $\mathbf{D}_{s_i}$ to norm unity (*Blumensath and Davies, 2009a*). This normalizes the eigenvalues of $\mathbf{D}_{s_i}^T \mathbf{D}_{s_i}$ to center near unity before calculating the eigendecomposition, allowing us to assess the degree to which the RIP is satisfied. This scaled matrix can be used directly in *Equation 22* (*Blumensath, 2013*; *Blumensath and Davies, 2009a*). The spectra of these matrices indicates that the RIP becomes far more weakly satisfied in the non-adaptive system than in the adaptive one, for sufficient odor complexity and intensity.

### Network model and classification

For the network model, the AL-to-MB connectivity matrix $\mathbf{J}_1$, is chosen such that each KC connects pre-synaptically to seven randomly chosen AL glomeruli (*Litwin-Kumar et al., 2017*; *Caron et al., 2013*). The results shown in *Figure 5* are an average of 10 distinct instantiations of this random topology. The $Z = 2500$ KCs are then connected by a matrix $\mathbf{J}_2$ to a readout layer of dimension $Q$, where $Q = 2$ for binary and $Q = N_{\text{ID}}$ for multi-class classification. Both AL-to-MB and MB-to-readout connections are perceptron-type with rectified-linear thresholds. The weights of $\mathbf{J}_1$ and $\mathbf{J}_2$ are chosen randomly from $\sim \mathcal{N}(0, 1/\sqrt{7})$ and $\sim \mathcal{N}(0, 1/\sqrt{Z})$, respectively. Only the $\mathbf{J}_2$ and the MB-to-output thresholds are updated during supervised network training, via logistic regression (for binary classification) or its higher-dimensional generalization, the softmax cross entropy (for multi-class classification).

## Acknowledgements

NK was supported by a postdoctoral fellowship through the Swartz Foundation and by an NRSA postdoctoral fellowship through the NIH BRAIN Initiative under award number 1F32MH118700. TE was supported by NIH R01 GM106189. We thank Damon Clark, John Carlson, Mahmut Demir, Srinivas Gorur-Shandilya, Henry Mattingly, and Ann Hermunstad for comments on the manuscript.

## Additional information

### Funding

| Funder | Grant reference number | Author |
| --- | --- | --- |
| Swartz Foundation | Postdoctoral Fellowship | Nirag Kadakia |
| National Institutes of Health | R01 GM106189 | Thierry Emonet |
| National Institute of Mental Health | F32 MH118700 | Nirag Kadakia |

The funders had no role in study design, data collection and interpretation, or the decision to submit the work for publication.

### Author contributions

Nirag Kadakia, Conceptualization, Software, Formal analysis, Investigation, Methodology, Writing—original draft, Writing—review and editing; Thierry Emonet, Conceptualization, Supervision, Methodology, Writing—review and editing

### Author ORCIDs

Nirag Kadakia https://orcid.org/0000-0001-9978-6450
Thierry Emonet https://orcid.org/0000-0002-6746-6564

### Decision letter and Author response

Decision letter https://doi.org/10.7554/eLife.45293.020
Author response https://doi.org/10.7554/eLife.45293.021

## Additional files

### Supplementary files

• Transparent reporting form

DOI: https://doi.org/10.7554/eLife.45293.017

### Data availability

All data generated or analysed during this study are included in the manuscript and supporting files. All software codes are available via GitHub (https://github.com/emonetlab/ORN-WL-gain-control, copy archived at https://github.com/elifesciences-publications/ORN-WL-gain-control).

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
