## [Decision Letter]

Thank you for submitting your article "Front-end Weber-Fechner gain control enhances the fidelity of combinatorial odor coding" for consideration by *eLife*. Your article has been reviewed by three peer reviewers, and the evaluation has been overseen by a Reviewing Editor and Catherine Dulac as the Senior Editor. The following individual involved in review of your submission has agreed to reveal their identity: Katherine I. Nagel (Reviewer #2).

The reviewers have discussed the reviews with one another and the Reviewing Editor has drafted this decision to help you prepare a revised submission.

The reviewers agreed that the work was of potential interest. They also agreed, however, that the presentation needs to be improved considerably. To be considered further, the paper needs to be revised carefully considering *eLife*'s broad readership. This includes unpacking details about technical aspects of the work – such as embedding and compressed sensing – and explaining the key mathematical concepts of the model in words and (better) in schematic figures. The terseness with which the results were presented made evaluating the work difficult. The individual reviews below contain specific suggestions and comments that should help identify where such changes are needed, but we would also encourage the authors to ask for comments from several non-olfactory colleagues, particularly about accessibility.

Individual reviews follow:

*Reviewer #1:*

This paper investigates the impact of Weber adaptation in olfactory receptor neurons on olfactory coding using a model based on past experimental work (described in the paper this one is linked too). The central question should be of general interest, and the approach taken in the paper seems appropriate. I struggled, however, with the way the work is presented and this left me unsure about the conclusions reached. I am not an expert in olfaction, but I suspect these struggles will be shared by many other potential readers.

Response dynamics: I was quite confused about the importance of differences in response dynamics of different ORNs. In places the text appears to state that differences in dynamics are small (e.g. Introduction), and in others that they are important (subsection “Model of ORN sensing repertoire”). Some of this may originate from responses of a single cell to multiple odors vs. responses of different cells. Nonetheless, the present version of the paper is confusing in this regard.

Figure 2 and embedding: The embedding process used in the analysis illustrated in Figure 2 is not explained in any detail – meaning that I could not interpret Figure 2. Later in the Discussion (third paragraph) this figure is referred to with respect to response dynamics – this was particularly unclear. This figure is critical to the paper, so must be explained in more detail.

Figure 3: The use of compressed sensing in the decoding analysis in this figure is unclear. Related to this point, it's not clear how an appropriate tolerance is chosen (subsection “Front-end adaptation enhances odor discrimination in complex environments”, second paragraph). The approach to decoding needs to be described in considerable more detail.

Discrimination in complex odor environments: it is not clear here why the background should be represented as static. I would have thought it would be subject to many of the same properties that make the signal dynamic. The role/importance of short term memory is also unclear.

It would be interesting to see how important ORN-specific adaptation is for the results presented, as compared to a mechanism that acted universally across all ORN responses.

Equation 1: The origin of this equation could get explained in more detail.

Equation 2: This form of feedback, and particularly its relation to Weber adaptation, should get explained more.

*Reviewer #2:*

This manuscript asks how adaptation in olfactory receptor neurons (ORNs) impacts the ability of an olfactory system to encode odor identities reliably. There is a broad consensus in the field that odors are encoded by the combinatorial activity of an array of receptors, each composed of an odor-specific receptor and a common co-receptor. At least one form of adaptation, in which the sensitivity of olfactory receptor neurons is adjusted based on the activation level of the receptor complex, is present within ORNs, likely acting at the level of feedback onto the orco co-receptor. This study uses theoretical approaches to ask how this form of adaptation impacts decoding of odor identity, using three different models of odor decoding: compressed sensing, primacy coding, and a biologically-inspired Kenyon cell model. The manuscript builds on a previous paper from the same group that developed a formulation for ORN adaptation based on a 2-state receptor model. The broad finding of the study is that front-end adaptation improves odor identity decoding using a variety of models. Overall I think this study addresses an important question and does so in a thorough way, making use of very reasonable models for both odor encoding and decoding, and providing a nice overview of the state of the field. However, I think some elements of the exposition could be made more accessible for less mathematically-inclined readers, and that some additional simulations would help pinpoint the reason why front-end adaptation improves encoding.

1) The manuscript is written for a highly quantitative audience and assumes a background familiar with the various models (receptor model, compressed sensing, t-SNE) they employ. I think the paper could be made more accessible by unpacking some of the mathematical formulae in the main text.

For example, it would be helpful to show a plot of the activation function A_a_ as a function of odor concentration (Equation 1) for some of their sample model neurons, in both the unadapted and adapted state.

In addition, the discussion of compressed sensing is highly compressed. If the authors could describe this in an intuitive or graphical way in the main Results it would help readers understand what this is and how it works.

Using a KC-inspired model to decode odor identity will probably be the most intuitive decoding scheme for many biologists. Here this decoding scheme is presented last but perhaps it might go earlier in the manuscript.

2) One possible interpretation of the results in Figures 2 and 3 is that in the non-adaptive system, high background odor concentrations cause the receptors to saturate, preventing them from encoding anything about the target odor, or at least massively compressing their dynamic range. This would mean that sensitivity adaptation is important (the activation curve needs to shift with increasing odor concentration), but not the precise form of the adaptation. Could the authors perform additional simulations to address this? For example: (1) What is the state of the receptors (distribution of activation levels) in the adapted versus un-adapted system in high background odor (prior to target odor presentation) vs. background+target? (2) How do the results in Figures 2 and 3 differ if the adaptation is not exact? That is, what if there is some factor 𝛽 in front of A_a_(t) in Equation 2? How precise does the adaptation have to be for this to work?

*Reviewer #3:*

The authors describe a receptor type-independent adaptation mechanism at the level of the olfactory sensory neurons (OSNs) that maintains odor capacity in natural conditions. They proposed that adaptation or gain control follows the Weber-Fechner Law of psychophysics (previously shown by the same group) and suggest that in a biological context it may be driven by Orco co-receptor activity in a non-receptor specific manner. The model results show that this kind of adaptation can aid concentration-invariant coding, discrimination (even in the presence of background odors) as well as it agrees with the novel hypothesis of primacy coding. The topic discussed in the article is relevant and the results are convincing, it is worth publishing; I have no major concerns.

[Editors' note: further revisions were requested prior to acceptance, as described below.]

Thank you for submitting your article "Front-end Weber-Fechner gain control enhances the fidelity of combinatorial odor coding" for consideration by *eLife*. Your article has been reviewed by one peer reviewer, Fred Rieke, who is the Reviewing Editor and Reviewer #1, and the evaluation has been overseen by Catherine Dulac as the Senior Editor.

*Reviewer #1:*

This is a revision of a paper describing a modeling approach to explore the role of front-end adaptation in olfactory coding. The paper is interesting, and contains a number of nice analyses that provide insight into the interaction between adaptation and coding. The revisions have made the paper easier to understand, but there are still several issues that are not as clear as they need to be. These, and some smaller points, are detailed below. In general (as detailed below), for each analysis I think it is essential that each of the steps involved in going from the modeled responses to a completed piece of analysis need to be clear to a non-expert reader.

1) t-SNE analysis.

The comparison of t-SNE to PCA (first paragraph of the subsection “Front-end Weber-Fechner adaptation preserves odor coding among background and intensity confounds”) is helpful. Can you build on the end of this paragraph to explain how t-SNE works, and, critically, to define the axes of Figures 2B, C? It is quite important that a reader is comfortable with what is being plotted here.

2) CS analysis.

The description of the sparseness constraint added to the paper is helpful. What is still not clear, however, is how the stimulus itself is estimated (I can guess, but it should be stated explicitly). Related to this point, the signal perturbation (Ds) is defined only in the Materials and methods but needed to interpret the main text (subsection “Front-end adaptation enhances odor decoding in complex environments”, third paragraph). It should also be clearer that you are decoding discrete odor identify, not concentration (assuming that is correct).

3) Decoding time-varying stimuli.

Several aspects of the analysis described in the fifth paragraph of the subsection “Front-end adaptation enhances odor decoding in complex environments” are not clear. Were odors randomly assigned to each whiff? And was the entire time course decoded, or was each whiff treated as a discrete event? In general, the description of this analysis needs to be considerably more detailed. Is there an intuitive argument as to why the longer time scales of adaptation are helpful that could be added to the last paragraph of the subsection “Front-end adaptation enhances odor decoding in complex environments”?

4) Tests of primacy coding.

The text suggests that the background odors may interfere with primacy coding in the absence of front-end adaptation. The analysis presented in Figure 4 then shows that primacy sets are maintained in the presence of front-end adaptation. But there is not a test, unless I missed it, of the initial suggestion that primacy sets are not maintained without front-end adaptation. This test is needed to interpret this section.

5) Interplay of front-end adaptation and divisive normalization.

Figure 5 suggests that these two forms of gain control may play quite different roles in coding discrete odor categories (aversive, appetitive) and odor identity. This is quite interesting. I would consider swapping the order of presentation so you start with a discussion of odor identify (that flows more naturally from the previous sections). The differences between the present results and those of Olsen et al. should also, at a minimum, get discussed in more detail.

---

## [Author Response]

Reviewer #1:This paper investigates the impact of Weber adaptation in olfactory receptor neurons on olfactory coding using a model based on past experimental work (described in the paper this one is linked too). The central question should be of general interest, and the approach taken in the paper seems appropriate. I struggled, however, with the way the work is presented and this left me unsure about the conclusions reached. I am not an expert in olfaction, but I suspect these struggles will be shared by many other potential readers.Response dynamics: I was quite confused about the importance of differences in response dynamics of different ORNs. In places the text appears to state that differences in dynamics are small (e.g. Introduction), and in others that they are important (subsection “Model of ORN sensing repertoire”). Some of this may originate from responses of a single cell to multiple odors vs. responses of different cells. Nonetheless, the present version of the paper is confusing in this regard.

Our use of the wording “response dynamics” was confusing because it did not distinguish between two key contributions to ORN response: 1) odor-receptor binding and activation of the OR-Orco complex (Equation 2); 2) signal transduction and adaptation (Equations 3-4). Because odor binding and activation is nonlinear, variability in Step 1 introduces variability in the dynamic response of the ORN, even though the filter used for the firing rate is assumed the same for all ORNs. We edited the text to make clear that it is the signal transduction and adaptation dynamics that exhibit a surprising degree of invariance with respect to odor-receptor identity, not the odor binding and ion channel activation (subsection “Model of ORN sensing repertoire”, sixth paragraph).

Figure 2 and embedding: The embedding process used in the analysis illustrated in Figure 2 is not explained in any detail – meaning that I could not interpret Figure 2. Later in the Discussion (third paragraph) this figure is referred to with respect to response dynamics – this was particularly unclear. This figure is critical to the paper, so must be explained in more detail.

We rewrote portion of the text to better explain how the embedding is enacted, and why we use t-SNE (versus PCA) to quantify the capability of the ORN repertoire to encode diverse odorants (subsection “Front-end Weber-Fechner adaptation preserves odor coding among background and intensity confounds”, first paragraph). We added a new panel A to Figure 2 to better introduce our approach and to help the reader interpret the other panels in the figure.

The later discussion about response dynamics is intended to draw parallels between our clustering results in Figure 2 and previous published results in which time traces of spiking activity were projected to a 3-dimensional space. In both cases, responses cluster by odor identity. In these studies the authors used the entire time trace, while here we consider the response at a single time. We amended the text in the Discussion to clarify this (fifth paragraph).

Figure 3: The use of compressed sensing in the decoding analysis in this figure is unclear. Related to this point, it's not clear how an appropriate tolerance is chosen (subsection “Front-end adaptation enhances odor discrimination in complex environments”, second paragraph). The approach to decoding needs to be described in considerable more detail.

Indeed, our discussion of compressed sensing (CS) was too terse. We changed the first panel of Figure 3 to make it more intuitive, removing the unnecessary equations and replacing them with a simple graphic. We also added text to describe more fully the general compressed sensing framework (subsection “Front-end adaptation enhances odor decoding in complex environments”, first two paragraphs).

We have explained the details for the decoding tolerance precisely in the Materials and methods, explaining our choice for this tolerance, and noting that our results are robust to particular choices in the tolerance.

Discrimination in complex odor environments: it is not clear here why the background should be represented as static. I would have thought it would be subject to many of the same properties that make the signal dynamic. The role/importance of short term memory is also unclear.

We are concerned with the detection of novel odors amid odors already present, thereby assuming that backgrounds odors have persisted for some time beforehand. Given that the adaptation time for the adaptation mechanisms we discuss is on the order of 250ms, the background of odor needs not to be strictly static. If it evolved on a slower time scale it would be enough. We chose to simplify the presentation so that one of these odors is on a much slower timescale, effectively static. This may be conceivable if the foreground and background arise from spatially separated sources: e.g. a lawn may release a background “grass odor” everywhere, while a flower in that lawn releases a foreground “flower odor” localized in plumes streaming from the flower. Of course, there are other cases where one odor of interest fluctuates on the same timescale as another nuisance odor. Then the distinction between foreground and background is lost. In our framework these would be considered both foreground odors.

The role of short term memory is to limit the amount of information utilized from the past. We now mention this in the fifth paragraph of the subsection “Front-end adaptation enhances odor decoding in complex environments”.

It would be interesting to see how important ORN-specific adaptation is for the results presented, as compared to a mechanism that acted universally across all ORN responses.

Indeed. Thank you for the suggestion. Please see response to reviewer 2 comment #2.

Equation 1: The origin of this equation could get explained in more detail.Equation 2: This form of feedback, and particularly its relation to Weber adaptation, should get explained more.

We have rewritten the text describing the model to provide more explanation and have added a step in the derivation of the former Equation (1) (now Equation 2) to make the derivation clearer. We have added paragraphs explaining the origin of Weber’s Law from the model, and two panels to Figure 1 to further illustrate the properties of the model and the Weber Law adaptation (subsection “Model of ORN sensing repertoire”;Figure 1F-1G).

Reviewer #2:[…] 1) The manuscript is written for a highly quantitative audience and assumes a background familiar with the various models (receptor model, compressed sensing, t-SNE) they employ. I think the paper could be made more accessible by unpacking some of the mathematical formulae in the main text.For example, it would be helpful to show a plot of the activation function A_a_ as a function of odor concentration (Equation 1) for some of their sample model neurons, in both the unadapted and adapted state.In addition, the discussion of compressed sensing is highly compressed. If the authors could describe this in an intuitive or graphical way in the main Results it would help readers understand what this is and how it works.

We have rewritten large parts of the paper to make this clearer. Please see our second, third and last responses to reviewer 1’s questions.

Using a KC-inspired model to decode odor identity will probably be the most intuitive decoding scheme for many biologists. Here this decoding scheme is presented last but perhaps it might go earlier in the manuscript.

We were also somewhat on the fence in the ordering of the results. We opted for this presentation mainly because primacy coding and compressed sensing decoding are more easily interpretable and far more tractable computationally without the added machinery of the AL and MB connectivity. Further, primacy coding has been shown in projection neurons, one step away from ORNs, so we presented it before we discuss the AL-MB connectivity. We do note in the CS section that we will later investigate the implications of circuit mechanisms in later sections. For these reasons, we chose to keep the ordering as is.

*2) One possible interpretation of the results in Figures 2 and 3 is that in the non-adaptive system, high background odor concentrations cause the receptors to saturate, preventing them from encoding anything about the target odor, or at least massively compressing their dynamic range. This would mean that sensitivity adaptation is important (the activation curve needs to shift with increasing odor concentration), but not the precise form of the adaptation. Could the authors perform additional simulations to address this? For example: (1) What is the state of the receptors (distribution of activation levels) in the adapted versus un-adapted system in high background odor (prior to target odor presentation) vs. background+target? (2) How do the results in figures 2 and 3 differ if the adaptation is not exact? That is, what if there is some factor* 𝛽 *in front of A_a_(t) in Equation 2? How precise does the adaptation have to be for this to work?*

Thank you for this suggestion. This comment (and reviewer #1 last comment) suggests a need to investigate how much we can break Weber scaling and still maintain combinatorial codes. We have now extended the section on odor coding to address this issue (subsection “Front-end Weber-Fechner adaptation preserves odor coding among background and intensity confounds”). We have introduced in our model a new parameter 𝛽 that allows us to gradually break the WeberFechner’s scaling. When 𝛽=0 Weber’s law is strictly satisfied and when 𝛽=1 there is no adaptation. Increasing 𝛽 away from zero introduces a dependency of the adapted state on the background odor concentration. We added two panels in Figure 2 illustrating this.

[Editors' note: further revisions were requested prior to acceptance, as described below.]

Reviewer #1:This is a revision of a paper describing a modeling approach to explore the role of front-end adaptation in olfactory coding. The paper is interesting, and contains a number of nice analyses that provide insight into the interaction between adaptation and coding. The revisions have made the paper easier to understand, but there are still several issues that are not as clear as they need to be. These, and some smaller points, are detailed below. In general (as detailed below), for each analysis I think it is essential that each of the steps involved in going from the modeled responses to a completed piece of analysis need to be clear to a non-expert reader.1) t-SNE analysis.The comparison of t-SNE to PCA (first paragraph of the subsection “Front-end Weber-Fechner adaptation preserves odor coding among background and intensity confounds”) is helpful. Can you build on the end of this paragraph to explain how t-SNE works, and, critically, to define the axes of Figures 2B, C? It is quite important that a reader is comfortable with what is being plotted here.

We have expanded our description of t-SNE as suggested. t-SNE only preserves local distances but not global distances. Thus, while it is a useful tool to cluster objects by similarity, the distances between clusters in the t-SNE projection are not necessarily meaningful because global distances are not preserved. We now mention this both in the main text and in the caption. We also explain in the main text that we use t-SNE primarily as a visualization method. To more rigorously quantify how well representations of odor identity are preserved, we calculate the mutual information (MI) between novel foreground odors and ORN responses in the 50-dimensional space. We also provide a short explanation of what MI is in the main text.

2) CS analysis.The description of the sparseness constraint added to the paper is helpful. What is still not clear, however, is how the stimulus itself is estimated (I can guess, but it should be stated explicitly). Related to this point, the signal perturbation (Ds) is defined only in the Materials and methods but needed to interpret the main text (subsection “Front-end adaptation enhances odor decoding in complex environments”, third paragraph). It should also be clearer that you are decoding discrete odor identify, not concentration (assuming that is correct).

We added explicit description of how the stimulus is estimated in the main text together with details about the definition of the signal perturbation. We also state clearly in the main text that the result is an estimate of the magnitude of each signal component and therefore both the identity and the intensity of the signal are estimated.

3) Decoding time-varying stimuli.Several aspects of the analysis described in the fifth paragraph of the subsection “Front-end adaptation enhances odor decoding in complex environments” are not clear. Were odors randomly assigned to each whiff? And was the entire time course decoded, or was each whiff treated as a discrete event? In general, the description of this analysis needs to be considerably more detailed. Is there an intuitive argument as to why the longer time scales of adaptation are helpful that could be added to the last paragraph of the subsection “Front-end adaptation enhances odor decoding in complex environments”?

We have revised the text at the location indicated in the review to address the reviewer’s concerns. Note that we hold the adaptation timescale at 250 ms, but vary the memory timescale (longer ones are more helpful). We also added a phrase in the following paragraph to provide intuition as to why longer time scales of memory are useful.

4) Tests of primacy coding.The text suggests that the background odors may interfere with primacy coding in the absence of front-end adaptation. The analysis presented in Figure 4 then shows that primacy sets are maintained in the presence of front-end adaptation. But there is not a test, unless I missed it, of the initial suggestion that primacy sets are not maintained without front-end adaptation. This test is needed to interpret this section.

We have now added on Figure 4C, the results for the non-adaptive case, showing that in absence of front-end adaptation, primacy sets are not well maintained, particularly when the primacy order is small.

5) Interplay of front-end adaptation and divisive normalization.Figure 5 suggests that these two forms of gain control may play quite different roles in coding discrete odor categories (aversive, appetitive) and odor identity. This is quite interesting. I would consider swapping the order of presentation so you start with a discussion of odor identify (that flows more naturally from the previous sections). The differences between the present results and those of Olsen et al. should also, at a minimum, get discussed in more detail.

This is an interesting observation. We have rewritten this part of the text as suggested and have mentioned in the main text the interesting observation made by the reviewer. We have also expanded the comparison of our results with those of Olsen et al.